# Non-zonal gravity wave forcing of the Northern Hemisphere winter circulation and effects on middle atmosphere dynamics

Sina Mehrdad<sup>1,2</sup>, Sajedeh Marjani<sup>1,3</sup>, Dörthe Handorf<sup>4</sup>, and Christoph Jacobi<sup>1</sup>

Correspondence: Sina Mehrdad (s.mehrdad@imperial.ac.uk)

Abstract. Gravity waves (GWs) are a major yet poorly constrained driver of middle-atmosphere dynamics. Using the high-top UA-ICON global circulation model, we conducted a set of six-member ensemble simulations in which orographic GW drag was selectively intensified over three Northern Hemisphere hotspots identified from observational and modeling studies—e.g., Himalayas (HI), Northwest America (NA), and East Asia (EA)—to assess their long-term dynamical impacts on the stratosphere. The imposed forcing generated distinctive vertical—horizontal drag structures in each region, yet produced a coherent hemispheric response. Resolved waves compensated the local drag through compensation mechanisms. In all three cases, added westward momentum suppressed upward and equatorward propagation of planetary waves, particularly of wavenumber 1, strengthening westerlies in the upper stratosphere—mesosphere. The frequency of sudden stratospheric warmings remained unchanged in the HI and NA experiments, but increased notably in EA, while the ratio of split to displacement events was unaffected. These results highlight the sensitivity of stratospheric variability to non-zonal GW forcing and underscore the importance of improving our understanding of GW-climate interactions. The simulation dataset presented here offers a valuable resource for future studies on gravity wave—induced variability in the climate system.

#### 1 Introduction

Internal gravity waves (GWs) are natural atmospheric oscillations that arise from the interplay of buoyancy and gravity as restoring forces. They play a crucial role in atmospheric dynamics by transporting energy and momentum from their sources to the regions where they dissipate (Andrews et al., 1987; Fritts and Alexander, 2003). GWs modulate vertical coupling between atmospheric layers (Yiğit and Medvedev, 2016). They contribute to large-scale circulation patterns, such as the summer-to-winter pole-to-pole circulation in the mesosphere (Lindzen, 1981), and are responsible for the formation of the cold summer mesopause (Nordberg et al., 1965; Björn, 1984). Additionally, GWs are key drivers of the Brewer–Dobson circulation (Alexander et al., 2010; Butchart, 2014), and major atmospheric oscillations, including the quasi-biennial oscillation (Holton and Lindzen, 1972), and the semiannual oscillation in the upper stratosphere and mesosphere (Baldwin et al., 2001).

<sup>&</sup>lt;sup>1</sup>Leipzig Institute for Meteorology, Leipzig University, Stephanstr. 3, 04103 Leipzig, Germany

<sup>&</sup>lt;sup>2</sup>Department of Earth Science and Engineering, Imperial College London, London, UK

<sup>&</sup>lt;sup>3</sup>Department of Physics, Imperial College London, London, UK

<sup>&</sup>lt;sup>4</sup>Alfred Wegener Institute for Polar and Marine Research, Research Unit Potsdam, D-14473 Potsdam, Telegrafenberg A43, Germany

Another major driver of middle atmospheric dynamics is planetary wave (PW) activity (Andrews and Mcintyre, 1976). When these waves break, they can disrupt the polar night jet and are widely recognized as the primary mechanism behind sudden stratospheric warmings (SSWs)—dramatic and rapid increases in polar stratospheric temperatures during winter, often accompanied by a reversal of the climatological westerly winds (Baldwin et al., 2021). GWs interact with PWs by modulating their behavior in the middle atmosphere. This interaction is often described through the compensation mechanism, whereby changes in GW forcing are offset by variations in resolved wave drag (Cohen et al., 2013, 2014; Sigmond and Shepherd, 2014; Karami et al., 2022). The impact of GWs on resolved waves strongly depends on the spatial and temporal distribution of the associated GW drag (Boos and Shaw, 2013; Shaw and Boos, 2012; Šácha et al., 2016; Kuchar et al., 2020), and particularly on the relative phase alignment of stationary planetary waves (SPWs) and the GW drag (Samtleben et al., 2019, 2020). GWs also play a role in both the initiation and evolution of SSWs (Baldwin et al., 2021), including their preconditioning, onset, and the subsequent recovery of the polar vortex (Richter et al., 2010; Limpasuvan et al., 2012; Albers and Birner, 2014; Šácha et al., 2016). These alterations in stratospheric dynamics can, in turn, influence weather patterns and climate in the troposphere through downward coupling mechanisms (Haynes et al., 1991; Baldwin and Dunkerton, 2001; Kidston et al., 2015).

GWs can originate from a variety of atmospheric processes, such as airflow over topography (Smith, 1980; Nastrom and Fritts, 1992), deep convection (Alexander et al., 1995), frontal systems and jet streams (Eckermann and Vincent, 1993; Plougonven and Zhang, 2014), and shear instabilities (Bühler et al., 1999). The propagation and eventual dissipation of GWs depend strongly on their source characteristics—such as phase speed and initial amplitude—and the surrounding atmospheric conditions. Depending on these factors, GWs may either propagate into the upper atmosphere or break at lower altitudes, depositing their energy and momentum locally (Gisinger et al., 2017). GWs break when they reach a nonlinear instability threshold, either due to amplitude saturation—where the wave's amplitude becomes so large that it induces local instabilities—or due to critical-level wave breaking, where the background wind creates a filtering layer. In a stably stratified, inviscid atmosphere, GW amplitudes typically increase exponentially with altitude because of the decrease in air density (Eliassen and Palm, 1961; Bühler et al., 1999). As amplitudes grow, they saturate and dissipate energy through the generation of turbulence. Amplitude saturation breaking refers to this process, where wave energy is dissipated via local instabilities in the flow (Lindzen, 1981; Fritts, 1984; Fritts and Alexander, 2003). In contrast, critical-level wave breaking occurs when a GW encounters a background flow whose horizontal wind speed matches or closely approaches the wave's horizontal phase speed (Fritts and Alexander, 2003; Alexander et al., 2010; Teixeira, 2014).

35

In current general circulation models (GCMs), such as the ICON (Icosahedral Nonhydrostatic) model (Zängl et al., 2015) in its global setup, a substantial portion of the GW spectrum comprises waves with spatial scales too small to be explicitly resolved. As a result, these subgrid-scale GWs must be parameterized (McLandress, 1998; Sandu et al., 2019; Kruse et al., 2022). Most GCMs implement two primary types of GW parameterization schemes: one for subgrid-scale orographic (SSO) waves and another for non-orographic (NO) waves. The characteristics of GWs generated by these sources differ substantially—particularly in their phase speeds, which strongly influence their ability to propagate into the middle and upper atmosphere (Andrews et al., 1987). Orographic GWs are produced by the flow over topography and are typically stationary relative to the surface, whereas non-orographic GWs originate from diverse other sources. These distinctions are explicitly

considered in the SSO and NO parameterization schemes used in ICON (Warner and McIntyre, 1996; Lott and Miller, 1997; Scinocca, 2003; McLandress and Scinocca, 2005). Consequently, the parameterized drag resulting from these two wave types exhibits distinct characteristics within the model, contributing differently to the overall momentum budget and vertical coupling processes.

GW activity exhibits substantial spatial variability and is distributed in a zonally asymmetric manner, with distinct hotspot regions where GW activity and drag is particularly pronounced (Ern et al., 2004; Fröhlich et al., 2007; Hoffmann et al., 2013; Schmidt et al., 2016). These hotspots are characterized by highly intermittent and variable forcing, primarily associated with prominent orography features such as the Himalayas and the North American Rockies (Hertzog et al., 2012; Hoffmann et al., 2013; Wright et al., 2013; Kuchar et al., 2020; Hozumi et al., 2024). In addition to these well-established orographic sources, observational evidence has revealed a notable northern hemisphere GW hotspot over East Asia, also primarily linked to orographic forcing (Šácha et al., 2015). Numerical modeling studies—including those employing specified dynamics and high-resolution global simulations—consistently reproduce these regions of enhanced GW activity, further confirming their robustness and climatological relevance (Gupta et al., 2024; Kuchar et al., 2020).

Previous studies have investigated the effects of GW hotspots on middle atmospheric dynamics using idealized simulations with constant GW drag forcing (Šácha et al., 2016; Samtleben et al., 2019, 2020), or through event-based composite analyses (Kuchar et al., 2020; Sacha et al., 2021). However, the long-term statistical climatic impacts of intensified and regionally localized GW forcing remain poorly understood. In this study, we use the upper-atmosphere extension of the ICON global model (Borchert et al., 2019) to explore the long-term climate effects of GW forcing in three Northern Hemisphere hotspot regions, namely the Himalayas (HI), Northwest America (NA), and East Asia (EA) under present-day climate conditions. We modify the SSO GW scheme to selectively intensify GW forcing in each hotspot region. The control and sensitivity simulations each include six 30-year-long run ensemble members. As emphasized by Samtleben et al. (2020) and Sacha et al. (2021), differences in assumptions about background atmospheric conditions can lead to discrepancies in the diagnosed impacts of localized GW forcing. By increasing the ensemble size, our setup better captures internal variability and enables a more robust assessment of the forced signal. Unlike previous studies that rely on idealized or temporally uniform forcing, our experimental design incorporates intermittent and variable intensified GW drag, providing a more realistic representation of atmospheric behavior. To the best of our knowledge, this is the first study to systematically investigate the long-term climatic implications of regional GW hotspot forcing using a high-top GCM with such a comprehensive setup.

# 2 Data and methodology

60

70

In this section, we describe the simulations, data, and analysis methods used in this paper.

# 2.1 Model simulations

We used the ICON general circulation model version 2.6.6 with upper-atmosphere extension (UA-ICON) to conduct our sensitivity simulations. ICON, developed jointly by the German Weather Service (DWD) and the Max Planck Institute for Me-

teorology, is formulated on an icosahedral-triangular grid and enables seamless simulations across scales ranging from global numerical weather prediction and large-eddy simulations to climatological timeframes (Zängl et al., 2015). The UA-ICON extension integrates an upper-atmosphere physics package and adapts the dynamical core from shallow- to deep-atmosphere dynamics (Borchert et al., 2019).

# 2.1.1 Model configuration






Using UA-ICON, we conducted a set of 30-year climate experiments in which sea surface temperature (SST), sea ice (SI), and greenhouse gases followed repeated annual cycles that represent the mean state of the current climate. All experiments employed the ICON R2B4 grid, providing an effective horizontal mesh size of approximately 160 km, with 120 vertical levels reaching altitudes of about 150 km, and a time step of 360 seconds. The ecRad radiation scheme was used in all simulations (Hogan and Bozzo, 2018; Rieger et al., 2019). The SST and SI fields were taken from the ERA5 (Hersbach et al., 2020) climatology over the 1979–2022 period, for which hourly data were averaged into a single annual cycle. This cycle was then applied recursively as the lower boundary condition, minimizing the influence of oceanic modes of climate variability, such as El Niño–Southern Oscillation and the Pacific Decadal Oscillation, by aligning them with their mean state in the current climate. For the greenhouse gases, we prescribed recursive, annually invariant concentrations of CO<sub>2</sub>, CH<sub>4</sub>, N<sub>2</sub>O, CFC-11, and CFC-12, derived by averaging the historical CMIP6 mixing ratios over 1979–2020 (Meinshausen et al., 2017). Atmospheric ozone concentrations were defined by blending data from the Global and regional Earth-system Monitoring using Satellite and in-situ data (GEMS) (Hollingsworth et al., 2008) and Monitoring Atmospheric Composition and Climate (MACC) (Inness et al., 2013) climatologies, both drawn from the Integrated Forecasting system (IFS) (European Centre for Medium-Range Weather Forecasts, 2010). As a result, our simulations used no transient forcing from the boundary conditions; all were driven by the same repeated seasonal cycles of SST, SI, and greenhouse gases.

In the ICON model, subgrid-scale GW drag is represented using two parameterization schemes: one for SSO GWs and another for NO GWs. The SSO parameterization scheme, which follows Lott and Miller (1997), applies subgrid orographic drag to the resolved horizontal wind when low-level flow encounters orography at the model grid scale. This drag generated by the scheme has two main components: low-level drag and GW drag at higher altitudes. The low-level drag results from flow blocking and wake effects, while the GW drag at higher altitudes arises from momentum flux deposition by upward-propagating GW waves that break via amplitude saturation or critical levels.

For the NO parameterization, ICON uses a spectral approach developed by Scinocca (2003) and McLandress and Scinocca (2005), based on the conceptual framework of Warner and McIntyre (1996). In the default configuration used in our model simulations, a fixed total momentum flux of approximately  $2.5 \times 10^{-3} \, \mathrm{N \, m^{-2}}$  per azimuth is launched uniformly in four horizontal directions and distributed across a phase-speed spectrum. These waves are initiated at a vertical level near 450 hPa and propagate upward. The dissipation of this flux, and therefore the drag on the mean flow, is then determined by local atmospheric conditions via wave saturation and critical-level absorption. This approach ensures a horizontally uniform source of wave momentum while still allowing the model's resolved temperature and winds to control where waves deposit momentum.

# 2.1.2 Experiment design and evaluation








To investigate the effect of GW drag hotspots in the middle atmosphere on middle atmosphere dynamics, we conducted a series of sensitivity simulations using regionally enhanced SSO drag in three hotspot regions. In these experiments, we modified the SSO parameterization so that whenever high-altitude GW drag is generated, it is amplified by a factor of 10 in designated hotspot areas, while the low-level drag remains unchanged to avoid introducing unintended forcing. The scaling amplifies only the magnitude of the high-altitude GW drag, preserving its direction as determined by the underlying SSO low-level wind-topography interactions. The factor of 10 was determined experimentally through preliminary test simulations using varying winter initial conditions, ensuring that the enhanced drag remains within the range of natural variability and preserves realistic dynamical forcing.

Figure 1 (left panel) presents an example of the SSO-induced zonal wind tendency at one model vertical level for a single time step in the unmodified scheme, illustrating notable spatial variability in both location and intensity. In this test case, the SSO tendency is modified and enhanced only for the pixel of interest highlighted in the left panel. The right panel of Figure 1 depicts the vertical profile of the SSO tendencies for this pixel of interest in both the standard and modified schemes, confirming that our adjustment to the scheme exclusively affects high-altitude drag and preserves low-level drag. Consequently, when this modification is applied to selected hotspot regions, the only transient difference between the control and sensitivity runs is the regionally enhanced high-altitude drag, with low-level drag remaining unchanged.

We conducted 4 sets of experiments: (1) a control run with no modifications (C), and (2–4) three sensitivity runs where the SSO drag was intensified by a factor of 10 over NA, HI, and EA. These regions have been recognized to be associated with GW hotspots in observational analyses (Hertzog et al., 2012; Hoffmann et al., 2013; Wright et al., 2013; Šácha et al., 2015) and are associated with major topographic features. Specifically, the EA domain spans  $30^{\circ} - 60^{\circ}N$  and  $110^{\circ} - 175^{\circ}E$ ; the NA domain spans  $30^{\circ} - 60^{\circ}N$  and  $100^{\circ} - 130^{\circ}W$ ; and the HI domain spans  $25^{\circ} - 45^{\circ}N$  and  $70^{\circ} - 100^{\circ}E$  in our experiments.

Wintertime stratospheric dynamics exhibit high internal variability (Sun et al., 2022), making ensemble analysis a useful approach for robust signal detection. To account for the internal variability and enhance confidence in detecting the forced signal, each experiment configuration (i.e., C, EA, NA, HI) was performed with 6 ensemble members generated by varying the initial conditions. For the start time of the first ensemble member in each experiment, the initial condition is the mean state on January 1 at 00:00 (averaged over 1979–2022) from the ERA5 data (Hersbach et al., 2020). For the subsequent five ensemble members, one specific year (1984, 1992, 2000, 2008, or 2016) was excluded from the mean each time. Each ensemble member run spanned 30 years, yielding 180 years of simulation data per experiment, except for the fifth ensemble member in the NA, which became numerically unstable after 15 years. Although increasing the number of ensemble members per experiment would further reduce bias from the internal variability, resource constraints limited this study to 6 ensemble members per experiment, which provides a reasonable balance between computational feasibility and robustness.

In our analysis, we discarded the first year of each simulation as the spin-up period and focused on data from the second year onward. Because our primary interest lies in Northern Hemisphere stratospheric wave activity, we examined only the extended winter season from November 1 to March 31 (NDJFM). Our analysis is based on ensemble means. For each set of sensitivity

simulations, anomalies were calculated as the difference between the ensemble mean of that set and the ensemble mean of the control simulation set. To assess the consistency of anomalies, we computed the same anomaly for each individual ensemble member. An anomaly was considered consistent if at least five out of six ensemble members exhibited the same sign as the ensemble mean anomaly. This criterion ensures that identified anomalies represent a robust response rather than ensemble variability. This consistency criterion applies to all figures and analyses presented in this study.

Figure 2 (left panels) shows the climatological SSO-induced U and V (zonal and meridional wind components, upper and lower row, respectively) tendencies in the control run, scaled by layer pressure and averaged over the 200–1 hPa (stratosphere) pressure levels. This scaling offers insight into the magnitude of the imposed forcing in each experiment. As expected, the control run exhibits stronger forcing over major topographic regions for both the U and V components. The three right columns of Figure 2 display the anomalies for EA, NA, and HI experiments (note the logarithmic color scale), showing consistently enhanced GW forcing over the hotspot regions as expected. In most cases, the forced anomalies share the same sign as the climatological values in the forced regions. The regional amplification introduces spatial discontinuities at hotspot boundaries; however, such sharp transitions are an inherent feature of the SSO scheme due to its discrete wave generation and breaking criteria. No unusual model stability issues were observed, apart from a partial instability in the NA fifth ensemble member, which did not affect the climatological results.

The horizontal wind response to the forcing is predominantly and consistently easterly and equatorward within the forcing regions (Figure 2). However, over the East Asian region (EA), the response is primarily poleward, with easterly forcing in the zonal direction over most of the hotspot area. For HI, westerly and predominantly northward SSO drag anomalies appear in the higher latitudinal band  $(40^{\circ} - 65^{\circ}N)$  outside the hotspot region. These anomalies are largely consistent. A similar but less consistent pattern is observed in NA for the same latitudinal band  $(40^{\circ} - 65^{\circ}N)$  outside the hotspot region. A similar pattern is also observed outside the forced region in EA, but it is mostly confined to central Siberia and the eastern part of Northwest America and, in general, is less consistent.




Figure 3 shows the vertical profiles of the SSO-induced horizontal wind tendencies in three hotspot regions (namely East Asia, Northwest America, and the Himalayas). In each panel, solid lines depict the difference in the wind tendencies between the forced region ("in") and its corresponding latitude bands outside the forced region ("out") in both the control and sensitivity simulations. Dashed lines show how the forced region's tendencies differ from the control simulation in that same region.

The East Asia hotspot (left panels of Figure 3), in the control run (blue lines), showed predominantly westerly and poleward wind tendency anomalies relative to the corresponding "out" region through much of the stratosphere and mesosphere, with a local easterly anomaly in the lower to mid-stratosphere. Applying intensified forcing in EA (orange lines) further strengthened these relative zonal distribution tendencies, making them easterly in the stratosphere and lowermost mesosphere but westerly in the upper mesosphere, and stronger northward in both the stratosphere and mesosphere. Comparing the East Asia hotspot in EA directly with its control counterpart (orange dashed) confirms enhanced easterly and northward tendency anomalies, peaking in the lower mid-stratosphere for the zonal component and with two peaks in the meridional component (lower stratosphere and stratopause).

For the Northwest America hotspot (middle panels of Figure 3), in the control run, the hotspot exhibited stronger easterly and equatorward tendency anomalies compared to the corresponding "out" region in the lower stratosphere but westerly and poleward tendencies in the middle-upper stratosphere (blue lines). With intensified forcing in NA (green lines), the hotspot exhibited more easterly tendency anomalies compared to the corresponding latitudinal band throughout the stratosphere and mesosphere, turning westerly only around the stratopause. Additionally, the lower stratosphere showed clear equatorward tendency anomalies, while the upper stratosphere and mesosphere displayed poleward tendencies. Relative to the control run (green dashed), the Northwest America hotspot horizontal SSO-induced wind tendency anomalies became more easterly in NA throughout the stratosphere and mesosphere, with equatorward tendency anomalies in the lower stratosphere transitioning to poleward in the upper stratosphere and lower mesosphere.

For the Himalayas hotspot (right panels of Figure 3), the control climatology favored easterly and equatorward tendency anomalies relative to the corresponding "out" region throughout the stratosphere and mesosphere (blue lines). The exerted forcing in HI (red lines) amplified the same vertical pattern, and comparing the tendency anomalies within the hotspot region in HI to the control run (red dashed) showed a consistent intensification. Unlike the EA hotspot, however, the HI and NA hotspots exhibit most of their enhanced wind tendency anomalies in the lower stratosphere. These patterns remained largely consistent across the ensembles.

Overall, the analysis of SSO-induced horizontal wind tendencies confirmed that the sensitivity runs successfully applied the non-zonal GW-induced forcing in each hotspot, intensifying the SSO forcing so that the patterns follow the control climatology but are stronger in the "in" region compared to the latitude-matched "out" region. We also note that for each of the three hotspots, the forcing anomalies are different in the control run, and consequently, the effect of enhanced forcing differs.

# 2.2 Data




We used the daily model output for the extended winter season (NDJFM) in our analysis. The daily data represent the average of all time-step values over each simulation day. Only data for the Northern Hemisphere (poleward of 0° latitude) were used.

Parameterized GW wind and temperature tendencies, derived from SSO and NO schemes, were analyzed alongside resolved wind and temperature fields. The responses of resolved waves were examined using Eliassen-Palm (EP) flux diagnostics (Andrews et al., 1987), with additional insights from finite-amplitude wave activity (FAWA) (Nakamura and Solomon, 2010). FAWA and EP flux divergence provide complementary views on wave—mean flow interactions. Although closely related, FAWA quantifies the waviness of the polar vortex, representing deviations from the equivalent zonal mean state (Nakamura and Solomon, 2010), while EP flux divergence measures resolved eddy forcing on the mean flow. FAWA is particularly valuable in the context of localized GW forcing, as it can highlight small-scale or intermittent potential vorticity (PV) perturbations, features often associated with GW forcing, that may not be captured in the EP flux divergence signal. Unlike linear diagnostics, FAWA remains well-defined even in regions with weak or reversed PV gradients and is more sensitive to GW-induced PV adjustments, such as vortex edge shifts or filamentary structures.

FAWA is defined using quasigeostrophic (QG) PV, computed from daily outputs of zonal/meridional winds (u, v), temperature (T), and density  $(\rho)$  on pressure levels. The QG PV (q) is given by:

# SSO U tendency (Model level ~ 17 km)

# North west America Pixel of interest A random time step in the test simulation -0.004 -0.002 0 (m/s²)

#### Vertical profile of SSO drag tendency for the pixel of interest

**Figure 1.** (Left) Zonal wind tendency (U) over northwest America, corresponding to the Rocky Mountains, generated by the SSO scheme at approximately 17 km altitude for a single time step in a test simulation. The highlighted pixel indicates the location for which the vertical profile on the right-hand side is plotted. (Right) The vertical profile of the SSO zonal wind tendency for the highlighted pixel, showing the scheme output without modification (green) and after enhancement of high-altitude effects (red). The vertical level of 17 km (corresponding to the map's vertical level in the left panel) is indicated by a horizontal dashed line.

$$q = f + \zeta + \frac{f}{\rho} \frac{\partial}{\partial z} \left[ \frac{\rho(\theta - \tilde{\theta})}{\partial \tilde{\theta} / \partial z} \right], \tag{1}$$

where f is the Coriolis parameter,  $\zeta$  is relative vorticity,  $\theta$  is potential temperature,  $\tilde{\theta}$  is its global mean, and derivatives use finite differences on geopotential heights (Nakamura and Solomon, 2010). For a given latitude  $\phi$  and height level z, FAWA, represented by  $A(\phi, z)$ , measures the areal displacement of PV contours away from zonal symmetry:

$$A(\phi, z) = \frac{1}{2\pi a \cos \phi} \left[ \iint_{q \ge Q(\phi, z)} q \, dS - \iint_{\pi/2 > \phi' \ge \phi} q \, dS \right],\tag{2}$$

where  $Q(\phi,z)$  is chosen so that the area poleward of the PV contour q=Q (the left-hand integral's domain) equals the area of the spherical cap poleward of  $\phi$  at that level (the right-hand integral's domain), a is Earth's radius, and the area element is  $dS=a^2\cos\phi\,d\lambda\,d\phi$  (with  $\lambda$  represents longitude). For a geometric interpretation and further details, see Nakamura and Solomon (2010, their Eq. (4) and Fig. 1). Under conservative conditions (adiabatic and frictionless), the FAWA tendency balances the (EP) flux divergence:

$$\frac{\partial A}{\partial t} = -\frac{1}{\rho} \nabla \cdot \mathbf{F} + N,\tag{3}$$

Figure 2. (Left panel) Climatology of the SSO-induced U tendency (zonal component, first row) and V tendency (meridional component, second row), scaled by layer pressure and averaged over the vertical extent from 200 to 1 hPa, based on the ensemble mean of the control run (C) during the extended winter season (NDJFM). For the U tendency (first row), positive values indicate westerly tendencies, and negative values indicate easterly tendencies. For the V tendency (second row), positive values indicate poleward tendencies, and negative values indicate equatorward tendencies. (Right three panels) Anomalies of the mean scaled SSO U tendency (first row) and V tendency (second row) for the sensitivity experiments (EA-C, NA-C, and HI-C, respectively) relative to the control run, calculated using ensemble means. Dotted areas indicate regions where the anomaly is considered consistent, meaning that at least five out of six ensemble members exhibit the same anomaly sign as the ensemble mean anomaly. The forced regions in each experiment are outlined in green. The color bars, shown on the right-hand side of the climatology and anomaly panels, represent values on a logarithmic scale.

where  $\mathbf{F}$  is the EP flux and N collects nonconservative sources and sinks (Nakamura and Solomon, 2010). This diagnostic relation links finite-amplitude eddy forcing to mean-flow adjustments in the transformed Eulerian-mean framework.

# 3 Results


This section begins with an analysis of the climatology of key variables in the control simulation. We then assess the impact of GW hotspots on middle atmosphere dynamics by comparing the mean states between the control and sensitivity simulations. As noted previously, all figures are based on the ensemble mean of each simulation category, with individual ensemble members also used for consistency checks.

# 3.1 Climatology of the control run

In this section, we analyze the climatology of the control run. Figure 4 presents the occurrence frequency of major SSWs and the subset of split events in the Northern Hemisphere, as simulated in the control run and three sensitivity experiments. Major

Figure 3. Vertical profiles of the SSO-induced wind tendency for zonal (U, top row) and meridional (V, bottom row) components during the extended winter period (NDJFM), calculated using ensemble means. Each column corresponds to a specific sensitivity simulation: EA (first column), NA (second column), and HI (third column). The solid blue lines represent the respective control run climatology, calculated as the difference between the "in" region (where the forcing is applied in the sensitivity simulation) and the "out" region (at the same latitude range as the forced region but outside of it), as shown in the inset maps on the second-row panels. These solid blue plots illustrate the zonal distribution climatology of SSO forcing across different hotspot regions. The solid orange, green, and red lines in columns 1, 2, and 3, respectively, represent the climatology calculated as the difference between the "in" and "out" regions in each sensitivity simulation. Dashed lines depict the difference between the sensitivity and control simulations for the forced region in each specific sensitivity case. Thickened segments of the plots indicate regions where the difference is considered consistent.

SSWs are identified using the World Meteorological Organization (WMO) criterion—simultaneous reversal of the zonal-mean zonal wind at 60°N and the temperature gradient between 60°N and 90°N at 10 hPa—as applied in Labitzke (1981) and Charlton and Polvani (2007). A split event is classified as one in which, during the period of wind and temperature gradient reversal, before vortex recovery, the 10 hPa geopotential height fields show at least two days with exactly two distinct low-pressure centers—indicative of a vortex split. In the control run (leftmost bars in Figure 4), the mean total SSW frequency is  $5.06 \pm 0.52$  events per decade. This slightly underestimates the approximately six events per decade reported in reanalysis for the current climate (Charlton and Polvani, 2007; Kim et al., 2017). The proportion of split events in the control run is slightly smaller than one-half of the total, consistent with reanalysis-based climatology (Charlton and Polvani, 2007). The SSW frequencies in the sensitivity experiments will be discussed below in section 3.2.1.

Figure 5 shows the climatological structure of the zonal-mean zonal wind, EP flux, and its divergence. The control run reproduces a realistic wintertime zonal wind climatology, featuring a midlatitude subtropical jet in the upper troposphere and a distinct middle atmosphere jet centered at higher latitudes in the stratosphere and mesosphere, consistent with observations and reanalysis-based studies (Randel et al., 2004). The EP flux vectors and their divergence indicate realistic upward and poleward


**Figure 4.** Mean occurrence frequency (per decade) of major SSWs (solid bars) and split SSWs (hatched bars) across different simulations: Control (C; blue), Himalayas (HI; red), Northwest America (NA; green), and East Asia (EA; orange). Values represent ensemble means over six members, with purple error bars indicating the inter-ensemble standard deviation. All statistics are calculated for the extended winter season (NDJFM).

resolved wave propagation, with the dominant contribution arising from planetary-scale waves, particularly zonal wavenumber 1 (Edmon Jr et al., 1980).



Figure 6 presents the zonal-mean climatology of zonal wind and temperature tendencies induced by the GW parameterization schemes. The zonal wind tendencies produced by both schemes are predominantly easterly, acting against the prevailing westerly zonal-mean winds. The NO GW-induced tendencies (middle panel of Figure 6) are approximately an order of magnitude stronger than those induced by the SSO scheme (left panel), consistent with results in Karami et al. (2022) and Kunze et al. (2025) for UA-ICON. The strongest tendencies from both schemes occur in the upper stratosphere and lower mesosphere at mid- to high latitudes, highlighting the momentum deposition due to amplitude saturation breaking in both schemes. In the NO scheme, the zonal wind tendencies extend more prominently into the mesosphere and span a broader latitudinal range. In contrast, the SSO-induced tendencies are more localized, primarily confined to the upper stratosphere and lower mesosphere. Additionally, the SSO tendencies show a secondary maximum in the lower stratosphere over midlatitudes, within the so-called valve layer (Kruse et al., 2016), indicating critical-level breaking of stationary waves. The temperature tendencies (right panel of Figure 6) are largely dominated by the NO scheme, with the most substantial warming occurring in the mesosphere.

Figure 7 shows the climatology of FAWA as a function of latitude and pressure level. Regions of higher FAWA values indicate larger resolved wave amplitudes, which are often associated with stronger wave generation or breaking, particularly in the presence of nonlinear wave dynamics. In contrast, areas with low FAWA values correspond to weaker wave disturbances. The control simulation reproduces key features of the FAWA climatology reported in reanalysis-based studies (Nakamura and

Figure 5. Climatology of the extended winter Eliassen-Palm (EP) flux (black vectors) and its divergence (color shading) in the control run for all resolved waves (upper left panel), wavenumber 1 (upper right), wavenumber 2 (lower left), and wavenumber 3 (lower right). The zonal-mean zonal wind climatology (contour lines in  $m.s^{-1}$ ) for the control run is also shown in the upper left panel. Solid contour lines indicate positive values, while dashed lines indicate negative values. Reference vectors for the EP flux are provided in the top right of each panel; note the different scales.

Solomon, 2010), including elevated FAWA in the upper troposphere, as well as in the mid- and high-latitude stratosphere and mesosphere.

**Figure 6.** Zonal-mean climatology of the tendencies induced by GW parameterization schemes. The left panel shows the SSO-induced zonal wind tendency, the middle panel shows the NO GW-induced zonal wind tendency, and the right panel displays the temperature tendency resulting from the combined effects of both schemes.

**Figure 7.** Climatology of finite-amplitude wave activity (FAWA) from the control run. The data represent the ensemble mean for the extended winter (NDJFM) average.

# 3.2 Sensitivity simulations

In this section, we analyze the sensitivity simulations to assess the atmospheric response to the intensified SSO-induced momentum tendencies over the selected hotspot regions, relative to the control run.

#### 280 3.2.1 SSWs


Across all sensitivity experiments, the proportion of split to displacement SSWs remained comparable to the control run, about one-third to one-half, with no observed systematic shift in event type under intensified GW forcing (Figure 4). However, frequencies varied by hotspot: HI and NA showed total SSW rates similar to the control run but with greater ensemble variability, particularly in displacement events. In contrast, EA exhibited the highest frequency, closely matching reanalysis estimates of approximately six events per decade (Charlton and Polvani, 2007; Kim et al., 2017). This increase occurred in four of the six ensemble members, but the difference from the control run did not exceed one standard deviation.

# 3.2.2 GW drag anomalies







Figure 8 presents anomalies in zonal-mean zonal wind tendencies induced by the GW parameterization schemes for each sensitivity simulation, relative to the control run shown in Figure 6. Note that both SSO and NO schemes provide easterly tendencies so that negative anomalies are related to larger tendencies, while positive anomalies mean weaker tendencies.

A coherent response emerged in parameterized GW drag across experiments, with enhanced easterly SSO tendencies in the forced latitudinal ranges and NO tendencies modulating mesospheric drag. In HI, SSO anomalies were strongest in the lower stratosphere and consistently extended upward, accompanied by enhanced NO drag in the high-latitude mesosphere and weakened drag in the midlatitude upper stratosphere (Figure 8 a–c). NA followed a similar pattern but with weaker SSO easterly tendency anomalies confined to the lower stratosphere, lacking the vertical extension seen in HI and accompanied by consistent upper-level westerly anomalies (Figure 8 d–f). EA exhibited broader, vertically extensive SSO easterly anomalies without a distinct lower-stratospheric peak in the forced latitudinal range, and dominant NO easterly anomalies in the mesosphere (Figure 8 g–i). In all cases, the combined GW tendency anomalies reflected SSO dominance in the lower stratosphere, NO influence in the mesosphere, and westerly anomalies poleward of the forcing region in the stratosphere dominated by the SSO contribution.

#### 3.2.3 Zonal-mean zonal wind anomalies

Figure 9 illustrates the zonal-mean zonal wind anomalies in the sensitivity simulations relative to the control run. Intensified forcing produced easterly anomalies in the upper troposphere and lower stratosphere near the forced latitudinal ranges, often extending poleward, contrasted with high-latitude westerly strengthening in the upper stratosphere and mesosphere. In HI, easterly anomalies within the forced band extended into higher latitudes, while strong, consistent westerly anomalies developed throughout the high-latitude stratosphere and mesosphere (Figure 9 left panel). NA exhibited a similar structure, with weaker easterly anomalies extending farther poleward and reduced consistency in high-latitude anomalies (Figure 9 middle panel). EA featured easterly anomalies over the northern portion of the forced band that reached the pole, while the southern portion showed consistent westerlies extending into the upper stratosphere and mesosphere toward higher latitudes (Figure 9 right panel).

#### 3.2.4 EP flux and wave propagation

Across all experiments, resolved waves partly offset the imposed GW drag through suppressed upward and equatorward propagation, primarily from wavenumber 1. This produced positive EP flux divergence anomalies, and hence weakened resolved wave drag, in the mid- and high-latitude stratosphere and mesosphere. In HI, suppression began in the midlatitude upper troposphere and extended consistently to the polar mesosphere. Wavenumber 1 dominated this response, while wavenumber 2 exerted a modest, opposing influence (Figure 10). NA showed a similar structure but with less mesospheric reach and negative EP flux divergence anomalies in the lower mesosphere. High-latitude anomalies were less consistent in NA, and wavenumber 1 again dominated, with minimal opposing effect from wavenumber 2 (Figure 11). EA exhibited consistent suppression from the

**Figure 8.** Zonal-mean zonal wind tendency anomalies for the SSO scheme (left column; panels a, d, g), NO scheme (middle column; panels b, e, h), and combined SSO + NO schemes (right column, panels c, f, i), calculated as the difference between each sensitivity simulation and the control ensemble mean. Results are shown for the HI (top row; a-c), NA (middle row; d-f), and EA (bottom row; g-i) sensitivity simulations, each based on the ensemble mean of six members. Dotted areas indicate regions where the anomalies are consistent across ensemble members. Cyan contours represent the climatology of the zonal-mean zonal wind tendency  $(m.s^{-1}.day^{-1})$  from the control run, with solid lines denoting positive values and dashed lines indicating negative values. The latitudinal extent of the forcing region within each simulation is indicated by green dashed lines.

upper troposphere to the mesosphere, primarily within the northern part of the forced latitudinal region and extending into high latitudes. In contrast, the southern part showed a modest but consistent increase in upward and equatorward flux, confined to the lower stratosphere. Here, wavenumber 1 remained dominant and wavenumber 2 reinforced the total response (Figure 12). Wavenumber 3 had minimal impact overall.


Figure 9. Anomalies in the zonal-mean zonal wind for the HI (left panel), NA (middle panel), and EA (right panel) simulations. Each anomaly is calculated as the difference between the ensemble mean of six members from the respective sensitivity simulation and the control run. Dotted areas indicate regions where the anomalies are consistent across ensemble members. Cyan contours mark the climatology of the zonal-mean zonal wind  $(m.s^{-1})$  in the control run. The latitudinal extent of the forcing region in each simulation is indicated by green dashed lines.

# 3.2.5 FAWA and temperature anomalies




Figure 13 presents anomalies in FAWA (left column), zonal-mean temperature (middle column), and temperature tendencies from the combined SSO and NO gravity wave schemes (right column) for the HI (top row), NA (middle row), and EA (bottom row) simulations, each shown as a difference from the control run.

FAWA and temperature anomalies generally reflected the forced GW drag patterns, with signals over the forced latitudes extending poleward in the upper troposphere and lower stratosphere. In HI and NA, positive FAWA in the lower stratosphere indicated enhanced wave activity, accompanied by warming in the same region. Across experiments, temperature anomalies featured cooling in the high-latitude stratosphere and warming in the mesosphere, consistent with positive GW temperature tendencies in the high-latitude mesosphere.

In HI, FAWA showed consistent positive anomalies in the midlatitude upper troposphere/lower stratosphere, extending to high latitudes, negative anomalies in the mid- to high-latitude stratosphere, and positive anomalies in the upper mesosphere. Corresponding temperature anomalies included mid- and high-latitude warming in the upper troposphere/lowermost stratosphere, cooling in the high-latitude stratosphere, and mesospheric warming (Figure 13 a–c). NA showed a similar structure but with weaker and less consistent anomalies, polar-extended positive FAWA, and inconsistent mesospheric temperature anomalies (Figure 13 d–f). EA differed from both HI and NA, lacking a strong positive FAWA signal in the upper troposphere/lower stratosphere of the forced band, instead showing patchy, modest, but consistent positive anomalies in the lower stratosphere. Temperature anomalies featured polar warming and midlatitude cooling from the upper troposphere into the lower stratosphere, cooling in the high-latitude stratosphere, and warming in the mesosphere (Figure 13 g–i). GW temperature tendencies in all cases followed the expected thermal response to the corresponding GW drag anomalies (see Figure 8 c,f,i).

# 3.2.6 SPWs at 100 hPa

SPW responses at 100 hPa varied by hotspot and were shaped by the phase alignment between the GW forcing and the climatological wave. HI showed the largest amplitude changes, while NA and EA responses were generally weaker. In HI, the hotspot region is mostly out of phase with the SPW1 climatology across much of the forced latitudinal band, producing consistent negative amplitude anomalies in the forced region with localized amplitude increases on the southern edge. SPW2 showed a similar negative pattern, with consistent amplitude enhancements at higher latitudes. SPW3 was largely in phase, yielding positive anomalies in most parts of the forced latitudinal band (Figure 14). NA exhibited small and less consistent SPW1 amplitude changes, reflecting the hotspot position near the SPW1 climatological phase transition, and SPW2 and SPW3 displayed dipole-like structures with predominant amplitude increases along the edges of the forced region (Figure 15). EA featured SPW1 amplitude reductions along the northern edge of the forced latitudinal band and at higher latitudes, and SPW2 showed consistent reductions in the forcing band and at higher latitudes, both in line with the region being out of phase with the SPW1 and SPW2 climatology. SPW3 had positive anomalies along the flanks of the forced region but negative anomalies at higher latitudes (Figure 16). Phase responses generally mirrored the amplitude patterns. Negative shifts occurred where the hotspot was out of phase with the climatology, while positive shifts appeared near the edges of the forced regions (Figures 14–16).

Figure 10. EP flux diagnostics anomalies, calculated as the difference between the HI and control ensemble means. The first column shows anomalies in EP flux divergence, the second and third columns show anomalies in the vertical and horizontal components of EP flux, respectively. Rows correspond to all resolved waves (top row; panels a-c), wavenumber 1 (second row; panels d-f), wavenumber 2 (third row; panels g-i), and wavenumber 3 (bottom row; panels j-l). Positive values in the vertical and meridional components represent upward and northward flux anomalies, respectively. Dotted areas indicate regions where the anomalies are consistent across ensemble members. Cyan contours represent the climatology of each respective field from the control run: units are  $m.s^{-1}.day^{-1}$  for EP flux divergence and  $m^3.s^{-2}.10^6$  for EP flux vector components. The latitudinal range of the imposed forcing is indicated by green dashed lines.

**Figure 11.** Similar to Figure 10, but for the NA simulation.

Figure 12. Similar to Figure 10, but for the EA simulation.

Figure 13. Anomalies in FAWA (left column), zonal-mean temperature (middle column), and zonal-mean temperature tendency from the combined GW parameterization schemes (right column) for the HI (top row; panels a-c), NA (middle row; panels d-f), and EA (bottom row; panels g-i) simulations. Anomalies are calculated as the difference between the ensemble mean of each sensitivity simulation and that of the control run. Dotted areas indicate regions where the anomalies are consistent across ensemble members. Cyan contours indicate the climatology of each respective field from the control run, in corresponding units. The latitudinal extent of the forcing region is indicated by green dashed lines.

Figure 14. Stationary planetary waves (SPWs) and their anomalies, calculated using zonal wind at the 100 hPa level for wavenumber 1 (left panel), wavenumber 2 (middle panel), and wavenumber 3 (right panel). In the top row, cyan contours show the SPW climatology from the control run ensemble mean, while color shading indicates anomalies in the HI simulation relative to the control run. Dotted areas highlight regions where anomalies are consistent across ensemble members. The bottom row shows the amplitude and phase differences between the HI simulation ensemble mean and the control ensemble mean as a function of latitude for each wavenumber. Solid black lines represent amplitude differences (left y-axis), and purple dashed lines represent phase differences (right y-axis). Bolded sections of each line indicate latitudes where the differences are consistent across ensemble members.

Figure 15. Similar to Figure 14, but for the NA experiment. Note the different scaling of the SPW anomalies.

Figure 16. Similar to Figure 14, but for the EA experiment. Note the different scaling of the SPW anomalies.

# 4 Discussion





# 4.1 Vertical structure of GW drag

In this study, we focus on the climate effects of intensified GW forcing in three hotspot regions. The sensitivity experiments enhanced stratospheric GW drag within each targeted region, though with distinct vertical and spatial structures (Figures 2, 3, and 8). In the HI and NA experiments, the strongest drag anomalies were concentrated in the lower stratosphere, close to the upper flank of the subtropical jet. While the HI imposed GW drag extended upward into the upper stratosphere and lower mesosphere in the forced region (Figure 8a), the NA experiment exhibited a reduction in easterly SSO wind tendency aloft (Figure 8d). This suggests that intensified GW drag in the NA region may have increased the likelihood of critical-level filtering of the orographic GW, limiting their vertical momentum flux and weakening the drag higher up. In contrast, the EA region exhibited a vertically extensive enhancement of SSO GW drag throughout the stratosphere (Figure 8g), likely associated with a more vertically extended critical layer in this region (Pisoft et al., 2018).

Beyond the hotspot regions, intensified drag coincided with consistent anomalies in SSO zonal wind tendency in the stratosphere. A common feature across all sensitivity experiments was the emergence of westerly GW wind tendency anomalies in mid to high latitudes, particularly over northern Eurasia (Figure 2). In the zonal-mean context, these westerly anomalies likely compensated for the intensified westward forcing in the hotspots, contributing to the westerly SSO zonal wind tendency anomalies observed north of the forced regions in the stratosphere across all the sensitivity experiments (Figure 8).

# 4.2 Dynamical compensation via resolved waves

In the HI experiment, a narrow, strong easterly SSO drag anomaly was centered on the upper flank of the subtropical jet (Figure 8a), just outside the winter surf-zone edge, where the meridional PV gradient is typically large and the basic state is baroclinically stable (McIntyre and Palmer, 1984). By steepening the PV gradient, this localized torque could push the flow past the Charney–Stern stability limit (Charney and Stern, 1962), triggering barotropic/baroclinic instability and the spontaneous emission of resolved waves. The onset of this instability appeared as a positive FAWA anomaly collocated with the SSO GW drag maximum (Figure 13a). The resulting resolved-wave torque, evident in the positive EP flux divergence (Figure 10), balanced the imposed westward torque, confirming the stability-constraint compensation described by Cohen et al. (2013, 2014).

The same easterly forcing in the HI experiment weakened the westerly zonal-mean zonal wind in the upper-troposphere and lower-stratosphere hotspot latitudinal band (Figure 9 left panel) and, via the residual downward motion it induces, warms the lower stratosphere immediately poleward of the forcing region (Figure 13b). Thermal-wind balance links this warm anomaly to the negative wind shear and hence to the weakened westerlies seen aloft.

A broadly similar pattern was observed in the NA experiment, though the ensemble mean response was not consistent across ensemble members. In the EA experiment, the SSO GW forcing was broader and extended through much of the stratospheric surf zone (Figure 8g), rather than being confined to the upper troposphere and lower stratosphere as in HI. Because the drag here acted inside the region already dominated by PW breaking, the response followed the PV-mixing-constraint pathway

(Cohen et al., 2014). The added easterly torque disturbed the locally homogenized potential vorticity field; PW breaking then weakened, producing a column-wide negative FAWA anomaly (Figure 13g) and an associated positive EP-flux divergence pattern that compensated the imposed drag (Figure 12). The intensified SSO drag weakened the westerlies on the northern flank of the forced region, which consistently extended poleward throughout the lower and middle stratosphere (Figure 9 right panel). The resulting meridional shear tilt was accompanied by adiabatic warming in the mid-latitude lower stratosphere, consistent with thermal-wind balance (Figure 13h). A narrow positive FAWA patch in the lower stratosphere marked the edge of the forcing where limited instability could still develop. However, throughout most of the column, the wave–flow interaction was suppression rather than generation, as expected when GW forcing is distributed throughout the surf zone.

The localized weakening of the westerly zonal-mean zonal wind within part of the latitudinal range of the imposed GW drag and its northward extension may be a key driver of a negative Arctic Oscillation tendency observed in all sensitivity experiments (see Figure A1). Beyond this localized response, the broader zonal-mean zonal wind responses in the sensitivity experiments were primarily modulated by alterations in resolved wave activity. In all sensitivity experiments, the imposed GW drag led to a suppression of resolved wave upward propagation, particularly for wavenumber 1, consistent with findings from Samtleben et al. (2020) and Sacha et al. (2021) (Figures 10-12). Both the upward and equatorward wave propagation were suppressed, resulting in a strengthening of westerlies in the high-latitude upper stratosphere and mesosphere across all sensitivity experiments. A similar pattern was also observed in the associated refractive index analysis (not shown here). The intensification of westerlies in these regions was associated with cooling in the polar stratosphere (Figure 13 second column). In the HI experiment, this response was particularly strong and extended through the high-latitude stratosphere and mesosphere. The NA experiment, however, exhibited greater ensemble variability, and these patterns were less consistent across the midand high-latitude stratosphere and mesosphere.

# 410 4.3 Influence on SPWs and phase alignment




Analysis of SPW amplitude changes highlighted the importance of the relative position of the GW hotspots with respect to the SPW climatological phase (Figures 14-16). The observed patterns in the sensitivity experiments were similar to the "Group 1" hotspot locations introduced in Samtleben et al. (2020), where the imposed GW hotspots were partly out of phase with SPW1. Among the experiments, the HI simulation exhibits the strongest impact on SPW amplitude. This result aligns with findings from Samtleben et al. (2019), who reported that GW forcing located at lower latitudes tends to exert a more pronounced influence on planetary waves. Notably, the HI experiment also has the largest imposed forcing among the three configurations (Figure 8 a), which likely contributes to its stronger influence on SPW characteristics.

For the NA hotspot, the region lies within the phase transition zone of SPW1. This positioning might partly explain the high ensemble variability in the NA response, as small shifts in SPW1 phase across ensembles could lead to substantially different interactions with the imposed GW forcing.

# 4.4 Implications for SSW occurrence and type





The number of major SSWs per decade in the HI and NA experiments was similar to that in the control run, suggesting that enhanced GW forcing in these hotspot regions did not significantly alter the frequency of major SSWs (Figure 4). The larger ensemble variability in displacement event frequency observed in HI and NA may be linked to the dominant wavenumber-1 planetary wave response to the imposed forcing. In contrast, the EA simulation showed a notable increase in SSW occurrence compared to the control run, highlighting the potential importance of GW forcing in this region for SSW generation. This increase might be attributed to the more vertically extended drag forcing in the stratosphere over the East Asia hotspot. These findings are consistent with White et al. (2018), who emphasized the role of orographic GW forcing over the Mongolian Mountains in SSW occurrence, and Sacha et al. (2021), who found that East Asian GW hotspots events are associated with deceleration of the polar night jet. Our results also show that the zonal mean wind deceleration in EA extended further poleward than in the other experiments (Figure 9 right panel). However, this increase in SSW frequency was not observed in all ensemble members, indicating that internal variability may still play a role in modulating the EA response. Across all sensitivity experiments, the proportion of split events remained between one-third and one-half of the total SSWs, indicating that intensified regional GW forcing did not substantially alter the type of SSW events.

#### 435 4.5 Role of NO GW drag and mesospheric circulation

While orographic GWs, being stationary, are particularly susceptible to critical-level breaking whenever winds slacken aloft, NO GWs more commonly undergo gradual saturation unless a particular wind shear creates a critical layer. A consistent signal across the sensitivity experiments was the enhancement of eastward NO wind tendencies in the high-latitude mesosphere, although the magnitude and spatial structure varied among the three experiments (middle column of Figure 8). In our model setup, the NO GW parameterization launches GWs in the four cardinal directions, spanning a range of frequencies with a prescribed constant distribution. This setup produces a spectrum of breaking altitudes, making the exerted NO drag highly sensitive to the background atmospheric state. One of the primary factors controlling the zonal component of NO drag is the zonal-mean zonal wind. In each sensitivity experiment, changes in the wind environment modulated how much of the launched GW spectrum was filtered at critical layers versus how much propagated vertically. For example, in the HI experiment, a weakening of the climatological westerlies in the midlatitude lower stratosphere (Figure 9 left panel) resulted in westerly NO wind tendency anomalies on the upper flank of the weakened wind region (Figure 8b). This indicated reduced filtering of westerly momentum at lower altitudes. Conversely, when the background westerlies strengthen—such as in the high-latitude stratosphere and mesosphere in the HI experiment—more of the eastward-propagating waves were filtered lower down, allowing the easterly drag to increase higher in the atmosphere. These patterns consistently emerged across the sensitivity experiments, reflecting the close coupling between NO GW drag and the evolving background wind fields.

The easterly NO GW wind tendencies and the associated positive temperature tendencies were observed in the high-latitude mesosphere across all sensitivity experiments (right column of Figure 13), likely causing a poleward and downward residual circulation in this region. This enhanced circulation could account for the higher mesospheric temperatures observed in

each experiment, despite the presence of stronger westerly winds at these latitudes (middle column of Figure 13). However, it is worth noting that the temperature anomalies were not consistent in the NA experiment. Among the three sensitivity experiments, HI exhibited the strongest warming in the high-latitude mesosphere. This occurred despite the strongest and most consistent increase in zonal-mean westerly winds in the same region (Figure 9 left panel). Notably, the HI experiment also showed the strongest easterly NO GW wind tendencies in the high-latitude mesosphere (Figure 8b), which appeared to dominate the temperature response. As discussed previously, this anomalous westward tendency was likely related to the enhanced westerly winds at lower altitudes, which filter more eastward-propagating waves and allow westward momentum to deposit higher in the atmosphere. This contrast highlights the important role of enhanced westward GW drag in driving mesospheric warming through dynamically induced residual circulation, rather than through direct thermal coupling with changes in the mean zonal wind.

# 5 Conclusions



485

In this study, we employed the UA-ICON high-top global model (Borchert et al., 2019) to investigate the long-term impacts of intensified orographic GW forcing in three hotspot regions—East Asia, Northwest America, and the Himalayas—on middle atmospheric dynamics. By selectively enhancing the SSO GW drag in each hotspot region through targeted modifications to the parameterization scheme, and using ensemble simulations to isolate the forced signal from internal variability, we systematically examined the resulting dynamical responses in the middle atmosphere.

The sensitivity experiments revealed consistent enhancements of easterly SSO GW wind tendencies within the targeted hotspot regions, though with distinct vertical and horizontal structures unique to each location. While the zonal-mean response near the hotspot's latitudinal range was primarily driven by the imposed SSO drag, broader circulation changes were dominated by the resolved wave response. These resolved wave responses generally acted to compensate the combined effects of SSO and NO GW momentum forcing.

The NO GW drag response was largely modulated by changes in the background zonal-mean zonal wind in the stratosphere and played a critical role in shaping the net parameterized GW momentum tendency. This net tendency, resulting from the combined effects of SSO and NO GW drag, influenced the background state in a way that shaped the resolved wave response described above. The compensation between resolved and parameterized momentum forcing occurred through different mechanisms: in the HI and NA experiments, the forcing induced local instabilities consistent with the stability-constraint compensation mechanism, while in the EA case, the drag acted within the stratospheric surf zone, disturbing the PV field and activating the PV-mixing compensation pathway.

A common feature across all experiments was the suppression of upward and equatorward propagation of resolved wave activity—particularly planetary wave 1—resulting in enhanced high-latitude westerlies in the upper stratosphere and mesosphere. The spatial patterns of these responses were hotspot-specific and primarily governed by the relative phase alignment between the hotspot forcing region and the phase of SPW1.

Regarding SSWs, the EA experiment exhibited an increased frequency of events, suggesting that enhanced GW forcing in this region may promote SSW occurrence. However, no systematic changes in the type of SSWs (i.e., split vs. displacement events) were observed across the sensitivity experiments. A more detailed investigation into the preconditioning processes leading to SSWs in the EA region is warranted to better understand the underlying mechanisms and is out of the scope of the current study.

Overall, this study underscores the complex interplay between localized GW drag enhancements and large-scale atmospheric dynamics. The compensation between resolved and parameterized momentum tendencies demonstrates the inherent self-regulation within the middle atmosphere. The dataset and findings presented here offer a valuable foundation for future research into the role of localized GW hotspots in modulating stratospheric variability and climate interactions.

Code and data availability. The daily output datasets for all six ensemble members of each of the four experiments used in this study, the Control run, and the HI (Himalayas), NA (Northwest America), and EA (East Asia) sensitivity experiments, are publicly available through the World Data Center for Climate (WDCC) at DKRZ under the project CC-LGWF (https://www.wdc-climate.de/ui/project?acronym= CC-LGWF). The corresponding DOIs are as follows: the Control run is available at https://doi.org/10.26050/WDCC/UAICON\_GW\_C (Mehrdad et al., 2025a), the HI experiment at https://doi.org/10.26050/WDCC/UAICON\_GW\_HI (Mehrdad et al., 2025c), the NA experiment at https://doi.org/10.26050/WDCC/UAICON\_GW\_NA (Mehrdad et al., 2025d), and the EA experiment at https://doi.org/10.26050/WDCC/UAICON\_GW\_EA (Mehrdad et al., 2025b).

# **Appendix A: Surface response**

490

In this appendix, we briefly present the surface-level responses in the sensitivity experiments. Figure A1 shows the climatology from the control run alongside the anomalies in mean sea level pressure (MSLP) and 850 hPa temperature ( $T_{850}$ ) for each sensitivity simulation. All three experiments exhibit increased MSLP in high latitudes and over the Arctic, indicative of a negative Arctic Oscillation tendency in response to the hotspot forcing. However, the spatial extent of this response varies. In HI and NA, positive MSLP anomalies extend toward the Aleutian region, while in the EA experiment, the higher pressure anomaly is concentrated over the central Arctic. The  $T_{850}$  response also differs across experiments. In the HI and NA simulations, lower temperatures are observed consistently over North America and parts of Eurasia, accompanied by consistent warming over the North Pacific. In contrast, the EA experiment exhibits a more spatially coherent cooling over the Arctic. In general, the surface responses in the HI and NA simulations are similar and deviate from the EA experiment. A detailed investigation of the mechanisms driving these surface anomalies through stratosphere—troposphere downward coupling is beyond the scope of this study and is left for future work.

Author contributions. The study was conceived by Sina Mehrdad (S.M.) and Christoph Jacobi (C.J.), with significant input from all authors.

Model simulations were carried out by S.M. and Sajedeh Marjani (S.Ma.), with guidance from C.J. The first draft of the manuscript was

Figure A1. (Left column) Climatology of mean sea level pressure (MSLP; top row) and 850 hPa temperature ( $T_{850}$ ; bottom row) during the extended winter season (NDJFM), based on the ensemble mean of the control simulation (C). (Right three columns) Anomalies of MSLP (top row) and  $T_{850}$  (bottom row) for the sensitivity experiments (EA-C, NA-C, and HI-C) relative to the control run, calculated using ensemble means. Dotted regions indicate areas where anomalies are consistent across ensemble members. The green outlines mark the locations of the imposed SSO GW forcing in each experiment.

written by S.M., with further development by C.J. All authors contributed through discussions, feedback during the study, and review of the manuscript.

Competing interests. The authors declare that no competing interests are present.

Acknowledgements. We gratefully acknowledge the funding by the Deutsche Forschungsgemeinschaft (DFG, German Research Foundation)

- Projektnummer 268020496 – TRR 172, within the Transregional Collaborative Research Center "ArctiC Amplification: Climate Relevant Atmospheric and SurfaCe Processes, and Feedback Mechanisms (AC)<sup>3</sup>". This work used resources of the Deutsches Klimarechenzentrum (DKRZ) granted by its Scientific Steering Committee (WLA) under project IDs bb1238 and bb1438. We thank Prof. Corwin Wright and an anonymous referee for their constructive comments and suggestions, and Dr. Jiarong Zhang for his evaluation of the revised manuscript.

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
