# Peer review of "Non-zonal gravity wave forcing of the Northern Hemisphere winter circulation and effects on middle atmosphere dynamics"

_EGUsphere, 2025_

## Author Comment (AC1)

We thank the referee for the insightful comments, which have helped us improve the manuscript. Below, we repeat the reviewer's remarks in red italics, and add our respective responses in normal text.

*The authors investigate the long-term impacts of intensified orographic gravity wave (GW) forcing in three hotspot regions—East Asia, Northwest America, and the Himalayas—on middle atmospheric dynamics using the UA-ICON high-top global model. The experiments reveal consistent enhancements of easterly subgrid-scale orographic (SSO) GW wind tendencies within the targeted hotspot regions. The non-orographic (NO) GW drag response is largely modulated by changes in the background zonal-mean zonal wind in the stratosphere and plays a critical role in shaping the net parameterized GW momentum tendency. In all three cases, the added westward momentum suppressed upward and equatorward propagation of planetary waves, strengthening westerlies in the upper stratosphere–mesosphere. Overall, I think this topic is interesting, and the manuscript is well written. I have only one minor comments.*

We sincerely thank the reviewer for the positive and constructive feedback, and for recognizing both the interest of the topic and the quality of the manuscript. We greatly appreciate the careful reading and the helpful suggestion, which has improved the clarity of our methods section.

*Minor Comments:*

*Section 2.2 Data: The associated discussions of finite-amplitude wave activity are not easily understandable. It is recommended to provide a more detailed explanation of the method, along with the formula to enhance clarity for the reader.*

We thank the reviewer for pointing out that our original description of the finite-amplitude wave activity (FAWA) diagnostic in Section 2.2 could be expanded for clarity. We agree that providing a more detailed explanation and the relevant formula would improve the manuscript and make this diagnostic easier to follow.

In the revised manuscript, we have expanded Section 2.2 to address this point (lines 215-236 of the revised manuscript). The new text more clearly distinguishes FAWA from Eliassen–Palm (EP) flux, explains why FAWA is particularly useful in the context of localized GW forcing, and outlines its theoretical basis and the conditions under which it is well defined. We have included the explicit formulation for deriving FAWA, including the quasi-geostrophic PV and FAWA equations, consistent with Nakamura and Solomon (2010). In addition, we now present the diagnostic relationship between FAWA tendencies and EP flux divergence. We believe these revisions address the reviewer's concern and will make the methodology clearer for readers who may be less familiar with FAWA.

References:

Nakamura, N. and Solomon, A.: Finite-amplitude wave activity and mean flow adjustments in the atmospheric general circulation. Part I: Quasigeostrophic theory and analysis, Journal of the Atmospheric Sciences, 67, 3967–3983, https://doi.org/10.1175/2010JAS3503.1, 2010

---

## Author Comment (AC2)

We thank Prof. Corwin Wright for his thoughtful and constructive review, which has provided valuable guidance for improving the clarity and structure of our manuscript. Below, we repeat the comments in red *italics* and present our responses in normal text.

*In this manuscript by Merhdad et al, the authors asses the impact of scaled-up parameterised orographic forcing on the middle-atmospheric dynamics of the UA-ICON global model running at a ~160km grid size under climatological (i.e. repeating-annual) conditions. They find coherent responses at a hemispheric level with resolved waves compensating for the locally-induced drag and strengthened upper-stratospheric westerlies due to suppressed planetary wave propagation.*

*The paper is an interesting study, and I concur with Reviewer 1 that it is interesting and well-written, and worth accepting for publication. I have slightly more questions than Reviewer 1 though, and do think a minor set of corrections before acceptance would help strengthen this interesting study and help it better find an audience.*

*My main issue here relates to the structure of the text. I found the first half clear and easy to read, but the back half was much less structured - in particular sections 3.2 and 4 were ~155 lines and ~100 lines long respectively without a break and hence quite hard to read without losing track of where I was. I would strongly suggest restructuring the material here to be easier to read as I think a lot of the potential audience will get lost in this section, and I include a few suggestions below.*

*The written English and figure design are generally of an excellent quality - I do include a list of typos etc that I spotted below, but this is much lower than most papers I review!*

We are grateful to Prof. Wright for his positive assessment of our work and for noting the scientific interest and quality of the manuscript. It is particularly encouraging to receive such feedback from an expert in the field. We also appreciate that his comments highlight a key aspect, the structure and readability of the later sections, which we agree is essential for ensuring that the study reaches and engages its intended audience. We believe that addressing these points has significantly improved the manuscript's readability and overall presentation.

*Scientific/formatting comments:*

*L003: how were the hotspots identified?*

Thank you for the comment. We have clarified in the abstract (line 3 in the revised manuscript) that the hotspots were identified based on prior observational and modelling studies (e.g., Hertzog et al., 2012; Hoffmann et al., 2013; Wright et al., 2013; Šácha et al., 2015), as described in more detail in the introduction.

*L125 (1): The sharpness of the modification spatially seems like it would introduce sharp discontinuities both horizontally and vertically - a real wave would likely extend over several surrounding pixels and would similarly not appear out of nowhere in the vertical. Does this lead to any unusual model response or stability issues in the region?*

We thank Prof. Wright for this insightful comment. The regional amplification of sub-scale orographic (SS0) gravity wave (GW) drag was applied to grid cells within the defined hotspot boundaries for each sensitivity experiment, which may introduce spatial discontinuities relative to the control simulation. However, these boundaries typically lie within areas of climatologically low SSO GW drag (Figure 2, left column). While Figure 2 shows sharp changes in SSO GW drag at the hotspot edges in the sensitivity simulations, such transitions are inherent to the unmodified SSO parameterization scheme (Lott and Miller, 1997) in both horizontal and vertical dimensions.

For instance, in the horizontal domain, the scheme computes upward momentum flux based on subgrid-scale orography and surface wind conditions at each grid point (grid spacing ~160 km in our UA-ICON setup). Sharp transitions commonly occur, such as at coastlines, where no upward momentum flux is generated over ocean pixels, but adjacent land pixels may produce significant flux depending on topography and low-level wind. The generated vertical momentum flux is then deposited at vertical levels where breaking criteria (e.g., amplitude saturation or critical-level filtering) are met, often resulting in abrupt drag deposition in neighboring grid cells.

Vertically, the scheme evaluates momentum flux deposition layer by layer from the surface upward to the model top. Deposition begins at the first layer satisfying the breaking criteria (typically in the upper troposphere or stratosphere), transitioning from zero to a potentially high value based on background conditions (e.g., stability and wind shear). Due to the continuity of vertical layers, deposition typically ramps up or down gradually if conditions evolve favorably, but the initial onset can be sharp. This process continues until the flux is fully dissipated.

Test simulations and the main experiments presented in the paper showed no significant model stability issues (e.g., CFL violations) attributable to these discontinuities, beyond a partial instability in the fifth ensemble member of the NA experiment, which did not impact the overall climatological results. This explanation has been added to Section 2.1.2 of the revised manuscript (lines 167-170 of the revised manuscript).

*L125 (2): Just to check - when you introduce the scaling, you don't change the direction at all? I assume not, but it wouldn't hurt to be explicit.*

We thank Prof. Wright for this clarification request. No modifications were made to the wave generation mechanism or the physical characteristics determined by the interaction between surface wind and topography in the SSO parameterization scheme. Therefore, the scaling amplifies only the magnitude of the high-altitude GW drag without altering its direction, which depends on the topography shape, orientation, and low-level wind as computed in the

unmodified scheme. This point has been made explicit in Section 2.1.2 of the revised manuscript (lines 126-128 of the revised manuscript).

*L145: What is the impact of this being analysed as a 30 year mean? Presumably it leads to values being very smooth everywhere whereas a typical year would inevitably have anisotropies - could this affect anything about the subsequent model evolution? And would it matter to your results if it did - I suspect any effect would vanish within the spinup year?*

We thank Prof. Wright for this question. Our simulations were designed to represent the current climate, using the 43-year mean state from ERA5 (1979–2022) as the basis for initial conditions, consistent with prescribed sea surface temperature, sea ice, and greenhouse gas concentrations as boundary conditions in our setup. To generate ensemble members and include internal variability, we perturbed the initial state by subtracting one specific year from the overall mean for each member. This method introduces small but physically consistent differences in temperature, wind, and other fields, rather than applying random noise, while maintaining a realistic large-scale structure.

The primary goal of this ensemble design was to isolate the robust signals related to the imposed SSO GW forcing from internal variability, which influences both GW generation and wave–mean flow interactions. While the smoothed 43-year mean reduces short-term anisotropies in the base state, the year-specific perturbations reintroduce variability that affects model evolution through chaotic amplification. In addition to internal variability, the initial condition itself can influence the model output; therefore, a measure is required to minimize its effect. To avoid residual spin-up effects, we discarded the first simulation year and based our analysis on years 2–30. The 1-year spin-up is standard in AMIP-like simulations to account for equilibration of the dynamical fields (Benedict et al., 2013; Zhang et al., 2021). Given our focus on middle-atmosphere dynamics, this approach minimizes any initial-state artifacts, and with the spin-up period removed, any influence of the initial state on our results is expected to be negligible. A dedicated sensitivity analysis to separate initialization effects from intrinsic variability could be valuable for understanding divergence in high-top models, but this is beyond the scope of the present study

*L143, 149: six ensemble \*members\* per experiment? Or six ensembles of multiple members each? I am genuinely a bit puzzled here, and they're quite different things! I think it's members from the context of the rest of the manuscript?*

Thank you for pointing out this ambiguity. We have clarified the terminology throughout the manuscript to consistently use "ensemble member(s)" when referring to the six individual runs performed for each experiment (mainly lines 143-152 of the revised manuscript).

*Figure 2: it looks like the values in the boxes on these plots are extremely highly saturated even with the log scales. Could the scale be extended further so we can se where the values are actually peaking?*

Thank you for this suggestion. In the revised manuscript, we have adjusted Figure 2 by extending the color bar range to reduce saturation and allow peak values to be more easily distinguished. In addition, we have slightly zoomed in on each subplot to improve the visibility of the plotted features.

> *L238: I'm quite surprised that the non-orographic tendencies are so much larger than the orographic ones - is this normal for parameterisations of this type, or is it a feature of the increased magnitude of the drag you have produced (and if so, would the normal orographic contribution be even lower)? It feels odd given that in observations orographic sources seem to absolutely dominate the GW activity we see at these altitudes, even when averaged over zonal means. This is a genuine question as I'm not a parameterisation scientist - it's possible that this is normal for these schemes? Given my confusion here, it might be useful to maybe put in a second row showing what the tendencies are in the unmodified control run to help those who are similarly not entirely familiar with what "normal" is in this context.*

Thank you for this insightful question, which highlights an important aspect of GW parameterizations. In the UA-ICON model, the SSO parameterization follows Lott and Miller (1997), while the non-orographic (NO) scheme is based on Scinocca (2003) and McLandress and Scinocca (2005), the same frameworks used in the ECMWF IFS model underlying ERA5 (ECMWF, 2016). Figure 6 in the manuscript presents the climatology from the unmodified control run, reflecting the default model settings. In this configuration, NO GW wind tendencies are roughly an order of magnitude larger than SSO tendencies, consistent with results reported by Kunze et al. (2025) and Karami et al. (2022) for UA-ICON.

This relative dominance of NO drag is typical for such parameterizations. The NO scheme uses a spectral approach that launches a constant upward momentum-flux spectrum near 450 hPa in four cardinal directions at every grid point, irrespective of local conditions. Deposition then occurs primarily in the upper stratosphere and mesosphere, depending on background wind and stability. In contrast, the SSO scheme generates intermittent, stationary waves only when low-level wind–topography interactions are favorable, with the flux magnitude and direction varying substantially and deposition often occurring lower in the atmosphere (middle/lower stratosphere). Due to the exponential decrease in atmospheric density with height, the lower-altitude SSO drag exerts a stronger effective forcing per unit tendency, despite smaller absolute values. These differences are detailed in section 2.1.1 of the manuscript.

From an observational perspective, orographic sources often dominate GW activity at stratospheric altitudes in zonal means. However, GW parameterizations in GCMs are not strictly observation-constrained but tuned to produce realistic mean circulation (Alexander et al., 2010; Karami et al., 2022). In the revised manuscript, we have added a clarifying phrase in Section 3.1 (lines 263-264 of the revised manuscript) that compares the resulting tendencies in our control run with those reported in similar studies.

*Section 3.2 is extremely long and undifferentiated (155 lines, spread over >12 pages when figures are included), and this made it quite a hard read - I bounced off it several times, and I think a lot of readers would. I think there are two options here - either to break up the material into subsections, e.g. by region or variable, or (perhaps better) to try and synthesise broad conclusions form the figures and talk about them rather than going into detail about each individual panel. The content is interesting, but the way it's presented makes it quite hard to absorb as a reader, which is a shame as a lot of work has gone into it.*

We greatly appreciate your suggestions, as addressing them has significantly enhanced the readability and accessibility of the paper, making it easier for readers to follow the key findings without losing the scientific depth. In response to your comments, we have made the following revisions:

We acknowledge that section 3.2 was previously long and undifferentiated, and we have adopted your recommendation to synthesize broad conclusions from the figures while focusing on the main patterns and commonalities across experiments, rather than detailing each individual panel. This has allowed us to condense the content while preserving essential insights. Additionally, to improve flow and prevent readers from "bouncing off," we have broken the section into subsections organized by variable (e.g., SSWs, GW drag anomalies, EP flux), providing a clearer structure without fragmenting the narrative excessively.

*Section 4 has the same problem - three pages and 100 lines of text going into quite a lot of detail but without a clear overarching structure. What might help in particular here might be to summarise the key finding from this material in a digestible way for the reader, perhaps as a schematic figure showing the key findings being discussed, or a summary paragraph flagging up the key findings. It's all very interesting, it's just quite hard to read due to the density and length without any breaks.*

We greatly appreciate your suggestions. We acknowledge that section 4 was previously dense and unbroken, and we have adopted your recommendation to introduce a clearer structure by dividing it into subsections with descriptive headings that highlight overarching themes and logical progression (e.g., vertical GW drag structure, dynamical compensation, SPW phase alignment). This creates natural breaks and signposts for the reader, improving flow while preserving the integrative discussion. The subsections effectively flag core ideas upfront before delving into details.

These changes have not only improved the paper's organization but also strengthened its overall impact.

*Typos etc*

[Figure]

*L173: Figure 3 "shows", not "represents", surely?*

*L223: you say you "follow the methodology of..." but then cite two papers - is it one of these or both?*

*L228: "slighly"*

*L283: "eastely"*

*L354: "schems"*

*L363: "similar than" -> "similar to" (preposition issue)*

*L479: "experimnets"*

Thank you for pointing out these typographical errors. They have been corrected in the revised manuscript.

References:

Alexander, M. J.: Gravity waves in the stratosphere. In *The stratosphere: Dynamics, transport, and chemistry* (pp. 109–121). American Geophysical Union. https://doi.org/10.1029/2009GM000864, 2010.

Benedict, J. J., Maloney, E. D., Sobel, A. H., Frierson, D. M., and Donner, L. J.: Tropical intraseasonal variability in version 3 of the GFDL545 atmosphere model, Journal of Climate, 26, 426–449, https://doi.org/10.1175/JCLI-D-12-00103.1, 2013.

ECMWF, 2016: IFS Documentation Cy41r2 – Part IV: Physical Processes. ECMWF, Reading, UK, 213 pp. Available at: https://www.ecmwf.int/sites/default/files/elibrary/2016/17117-part-iv-physical-processes.pdf

Hertzog, A., Alexander, M. J., and Plougonven, R.: On the intermittency of gravity wave momentum flux in the stratosphere, Journal of the Atmospheric Sciences, 69, 3433–3448, https://doi.org/10.1175/JAS-D-12-09.1, 2012.

Hoffmann, L., Xue, X., and Alexander, M.: A global view of stratospheric gravity wave hotspots located with Atmospheric Infrared Sounder observations, Journal of Geophysical Research: Atmospheres, 118, 416–434, https://doi.org/10.1029/2012JD018658, 2013.

Karami, K., Mehrdad, S., and Jacobi, C.: Response of the resolved planetary wave activity and amplitude to turned off gravity waves in the UA-ICON general circulation model, Journal of Atmospheric and Solar Terrestrial Physics, 241, 105 967, https://doi.org/10.1016/j.jastp.2022.105967, 2022. Kunze, M., Zülicke, C., Siddiqui, T. A., Stephan, C. C., Yamazaki, Y., Stolle, C., Borchert, S., and Schmidt, H.: UA-ICON with the NWP physics package (version ua-icon-2.1): mean state and variability of the middle atmosphere, Geosci. Model Dev., 18, 3359–3385, https://doi.org/10.5194/gmd-18-3359-2025, 2025.

Lott, F. and Miller, M. J.: A new subgrid-scale orographic drag parametrization: Its formulation and testing, Quarterly Journal of the Royal Meteorological Society, 123, 101–127, https://doi.org/10.1002/qj.49712353704, 1997.

McLandress, C. and Scinocca, J. F.: The GCM response to current parameterizations of nonorographic gravity wave drag, Journal of the atmospheric sciences, 62, 2394–2413, https://doi.org/10.1175/JAS3483.1, 2005.

Šácha, P., Kuchaˇr, A., Jacobi, C., and Pišoft, P.: Enhanced internal gravity wave activity and breaking over the northeastern Pacific–eastern Asian region, Atmospheric Chemistry and Physics, 15, 13 097–13 112, https://doi.org/10.5194/acp-15-13097-2015, 2015.

Scinocca, J. F.: An accurate spectral nonorographic gravity wave drag parameterization for general circulation models, Journal of the Atmospheric Sciences, 60, 667–682, https://doi.org/10.1175/1520 0469(2003)060<0667:AASNGW>2.0.CO;2, 2003.

Wright, C., Osprey, S., and Gille, J.: Global observations of gravity wave intermittency and its impact on the observed momentum flux morphology, Journal of Geophysical Research: Atmospheres, 118, 10–980, https://doi.org/10.1002/jgrd.50869, 2013.

Zhang, Y., Yu, R., Li, J., Li, X., Rong, X., Peng, X., and Zhou, Y.: AMIP simulations of a global model for unified weather-climate forecast: Understanding precipitation characteristics and sensitivity over East Asia, Journal of Advances in Modeling Earth Systems, 13, e2021MS002 592, https://doi.org/10.1029/2021MS002592, 2021.